# Effects of Non-Essential “Toxic” Trace Elements on Pregnancy Outcomes: A Narrative Overview of Recent Literature Syntheses

**DOI:** 10.3390/ijerph20085536

**Published:** 2023-04-17

**Authors:** Maria Dettwiler, Angela C. Flynn, Jessica Rigutto-Farebrother

**Affiliations:** 1Human Nutrition Laboratory, Institute for Food, Nutrition and Health, ETH Zürich, 8092 Zürich, Switzerland; 2Department of Nutritional Sciences, King’s College London, London SE1 9NH, UK

**Keywords:** arsenic, cadmium, lead, mercury, pre-eclampsia, preterm birth, birth weight, birth length, head circumference

## Abstract

Adverse pregnancy outcomes and their complications cause increased maternal and neonatal morbidity and mortality and contribute considerably to the global burden of disease. In the last two decades, numerous narrative and systematic reviews have emerged assessing non-essential, potentially harmful, trace element exposure as a potential risk factor. This narrative review summarizes the recent literature covering associations between exposure to cadmium, lead, arsenic, and mercury and pregnancy outcomes and highlights common limitations of existing evidence that may hinder decision-making within public health. Several initial scoping searches informed our review, and we searched PubMed (latest date July 2022) for the literature published within the last five years reporting on cadmium, lead, arsenic, or mercury and pre-eclampsia, preterm birth, or prenatal growth. Pre-eclampsia may be associated with cadmium and strongly associated with lead exposure, and exposure to these metals may increase risk of preterm birth. Many reviews have observed cadmium to be negatively associated with birth weight. Additionally, lead and arsenic exposure may be negatively associated with birth weight, with arsenic exposure also adversely affecting birth length and head circumference. These findings should be interpreted with caution due to the limitations of the reviews summarized in this paper, including high heterogeneity due to different exposure assessment methods, study designs, and timing of sampling. Other common limitations were the low quality of the included studies, differences in confounding variables, the low number of studies, and small sample sizes.

## 1. Introduction

Adverse pregnancy outcomes are multifactorial [1], and the aetiology is poorly understood for both pre-eclampsia [2] and preterm birth [3], which are two of the leading causes of morbidity, though exposure to non-essential and potentially harmful “toxic” trace elements may be implicated. In 2019, McKeating, Fisher, and Perkins summarised the association between trace elements and gestational disorders. However, they focused only on the trace elements essential for health [4]. Several reviews have also been published on non-essential trace elements, including some that featured some pregnancy outcomes for selected non-essential trace elements [5,6,7] and the endocrine-disrupting properties of some trace metals and related infant and maternal outcomes [8]. To date, data on non-essential metal intake and gestational disorders are inconsistent, and despite further studies that have been published over the last five years, knowledge gaps remain. This narrative review, therefore, focuses on summarising the most recent literature covering the association between non-essential and potentially harmful trace elements, and pregnancy outcomes of pre-eclampsia, premature birth, and prenatal growth. We outline how prenatal exposure to these trace elements might take place, how exposure is commonly measured in this field of research, and the effect these elements might have on the pregnancy outcomes described above. Finally, a discussion provides a summary of the limitations cited in the referenced literature to inform the reader of the complexities and challenges associated with research in this important area.

## 2. Methods

After conducting several initial searches to inform our study question over the period of October to November 2021, we found that much of the relevant literature focuses on cadmium, lead, arsenic, and mercury exposure during pregnancy; thus, our focus rests on those four elements. Though older reviews in this area were found, since a number of articles have been published more recently and, would hence not be included in these older reviews, we chose to limit our synthesis to the literature published in the last five years. We included both the systematic and the narrative literature reviews that included human studies. Animal data are beyond the scope of this paper. Searches for the relevant literature were performed through PubMed in January of 2022 and again in July of 2022. To identify relevant manuscripts, the four elements, that is, cadmium, lead, arsenic, and mercury, were combined by using the OR function and then connected to different outcome terms by using the AND function. Specific search terms for pregnancy outcomes were pre-eclampsia, preterm birth, and prenatal growth. “Preterm delivery” and “premature birth” were also used as synonyms to preterm birth, and for prenatal growth, we also used “birth outcomes”, “birth size”, “birth weight”, “birth length”, “head circumference”, “fetal growth”, “prenatal growth”, “low-birth-weight”, “small-for-gestational-age”, and “intra-uterine growth restriction”. In addition, we applied the PubMed filters for the document types “Meta-Analysis”, “Review”, “Systematic Review”, and “Books and Documents”. The [AllFields] filter was applied throughout. In addition, human trials cited in reviews that are of particular reference herein have been discussed, though no explicit search was conducted for primary studies. Relevant manuscripts were selected, and we present them in a narrative synthesis in the following paragraphs.

## 3. Prenatal Exposure to Non-Essential (Toxic) Trace Elements

Exposure to cadmium, lead, arsenic, and mercury is well known to cause adverse health effects in humans. Humans are commonly exposed via skin contact, inhalation, or ingestion of contaminated food and drinking water [9], and the different exposure sources can be categorized into occupational, behavioural, nutritional, and environmental exposure, as visualised in Figure 1.

### 3.1. Cadmium

Cadmium is a heavy metal ubiquitous in the environment. It is naturally occurring in the crust of the earth and emitted in soil, water, and air due to anthropogenic activities, such as non-ferrous metal mining and refining and processes involving phosphate fertilisers, fossil fuels, and waste. Cadmium is also used for pigments, coatings, plating, and stabilisers for plastics. It strongly binds organic matter and thereby leads to exposure through the food supply and tobacco leaves in smokers [12]. After absorption, cadmium diffuses through the blood to various organs, notably the kidneys and liver, where it can cause damage [13,14,15]. The mechanisms of cadmium toxicity might involve the synthesis of metallothioneins, induction of oxidative stress, and epigenetic changes [14]. Several epidemiological and experimental studies suggest that chronic cadmium exposure can be associated with different types of cancer [14,16,17]. However, there are reviews on cadmium exposure and cancer types that cannot rule out a possible association but highlight many research gaps and inconsistent findings [18,19,20,21].

During pregnancy, cadmium diffuses to the placenta, where it accumulates [10,11]. However, it seems that cadmium does not cross the placental barrier easily. According to a study conducted in Sweden, cord blood cadmium was measured to be only 10% of the total cadmium concentration in maternal blood [22]. This study was also included in a review that assessed the cadmium levels of five studies. This review found cadmium cord blood levels to always be far below those in maternal blood, suggesting that the placenta acts as a partial barrier for cadmium. However, the authors concluded that due to the cadmium storage in the placenta and its consequences on placental function, this also poses a threat [10]. Moreover, the adverse effect cadmium has on the placenta and embryo has also been described due to its effect on gene methylation [23]. The efficiency of the placental barrier is thought to be modulated by specific maternal gene polymorphisms. Further studies on this topic are required [24,25].

### 3.2. Lead

Like cadmium, lead is also found in the crust of the earth and is emitted due to anthropogenic activities [9]. Humans are exposed to lead via inhalation, ingestion, and skin contact, though through the dermal route, lead poisoning is unlikely [26]. Due to its toxicity, lead is being phased out of gasoline worldwide [27,28] and reduced in many other areas of previous use. However, exposure is still common, especially in areas close to mining areas [29], landfills, and industrial or hazardous waste site. Lead is also used as a pigment and in storage batteries in vehicles [26,30]. Lead exposure can cause damage to many organs and systems affecting, namely, the neurological and vascular systems and renal function, as well as reproduction [31,32].

To examine the diffusion and accumulation of lead during pregnancy, Singh et al. compared metal concentrations measured in the placenta with pollution levels in several countries and found a positive correlation, including in cord blood, since lead easily crosses the placenta via passive diffusion [15]. Gundacker et al. assessed lead levels of five studies and observed that the levels of cord blood were equal to or lower than that of maternal blood [10]. This is consistent with the systematic review from Esteban-Vasallo et al. They concluded that lead passes the placental barrier without accumulating in the placenta [11]. Consequently, the US Centers for Disease Control and Prevention recommend interventions and follow-up in pregnant women with blood lead levels of 5.0 µg/dL and higher [33].

### 3.3. Arsenic

Arsenic occurs in many organic and inorganic forms [34]. Organic arsenic is considered less toxic and is present in surface water [35] and seafood [34,36]. In contrast, inorganic arsenic is a highly toxic metalloid and is present in groundwater and soil [35,36,37]. Humans are primarily exposed to arsenic through drinking water, as well as through food [34,37]. Arsenic exposure has been associated with short- and long-term health effects such as cardiovascular diseases, adverse effects on the immune system and respiratory infections [37,38,39], and several cancer types [37,38,40]. Suggested mechanisms include the induction of oxidative stress and inflammation [37,39], as well as genomic instability and epigenetic regulation [38,40].

During pregnancy, arsenic accumulates in the placenta, and both inorganic arsenic and methylated metabolites easily cross the placental barrier. Improved arsenic methylation and urinary excretion seem to provide partial protection against arsenic-induced toxicity during pregnancy [41].

### 3.4. Mercury

Mercury exists in elemental, organic, and inorganic forms. In contrast to arsenic, both inorganic and organic mercury are extremely toxic to humans [15]. Three important exposure routes of mercury exposure are fish consumption due to methyl mercury in fish, ethyl mercury in vaccines, and occupational exposure such as mercury vapor from amalgam tooth fillings. There are anthropogenic sources of mercury, such as gold mining, and natural sources such as volcanoes [42]. The different forms of mercury have different toxicity profiles, resulting in a broad range of potential symptoms [43,44]. Mercury exposure can also be categorised into occupational exposure and non-occupational exposure. Gold mining and fish/shellfish consumption play a role in both categories and, therefore, deserve special attention in the process of limiting mercury exposure [45].

The most common form of organic mercury that humans are exposed to is methylmercury. When pregnant women are exposed to it, mercuric ions may accumulate in placental and fetal tissues [46]. Esteban-Vasallo et al. reported inconsistent findings, though some studies suggested placental accumulation [11], and usually, cord blood mercury levels are higher than those in the maternal circulation [10].

## 4. Effects of Non-Essential Trace Elements on Pregnancy Outcomes

### 4.1. Pre-Eclampsia

Pre-eclampsia belongs to the hypertensive disorders of pregnancy and is characterised by the onset of hypertension during pregnancy and, often, the onset of proteinuria at or after 20 weeks of gestation. The American College of Obstetricians and Gynecologists revised their definition of pre-eclampsia in 2013 such that proteinuria is no longer required for diagnosis, but hypertension can also be accompanied by new-onset thrombocytopenia, impaired liver function, renal insufficiency, pulmonary edema, or visual or cerebral disturbances [47]. Pre-eclampsia is a significant public health concern and causes serious maternal and fetal morbidity and mortality [47,48]. A cohort study by Xiao et al. examined the effect of pre-eclampsia on fetal growth and found a 3.6-fold (95% CI, 2.3 to 5.7) increased risk to deliver a small-for-gestational-age infant and a 3.8-fold (95% CI, 1.9 to 7.5) increased risk of low birth weight [49].

In 2018, Rosen, Muñoz et al. published a systematic review assessing several environmental contaminants. They included seven studies measuring cadmium levels and concluded that there is strong evidence for an association between cadmium and the development of pre-eclampsia [48].

The only meta-analysis including pre-eclampsia as an outcome and identified in this review examined blood lead concentrations and pre-eclampsia, revealing a strong association with an increment of 1 μg/dL blood lead level, corresponding to a 1.6% increase in the likelihood of pre-eclampsia (odds ratio (OR) 9.81; 95% CI 8.01 to 12.02). The authors concluded that lead exposure is a major, if not the biggest, risk factor for pre-eclampsia, and cite studies that support a causal pathway. They not only suggest that pregnant women should be advised to avoid lead exposure but also suggest routine testing of blood lead concentration in pregnant women with historical lead exposure, as well as active monitoring and calcium supplementation for pre-eclampsia in women with high blood lead concentrations [50]. Another review included three additional studies, two of which supported the relationship between lead exposure and pre-eclampsia, though a case–control study did not observe an association, possibly due to the lack of adjustment for any covariates performed in the analysis of that specific study [51].

Regarding arsenic, Rosen, Muñoz et al. did not find an association with pre-eclampsia. However, they report that data were limited since their conclusion was based on only three studies, and one of the two studies that did not find an association hypothesised that their observation was due to a low concentration in the study population [48]. Based on the same three studies, Kahn and Trasande also concluded that there does not seem to be an association [51].

The systematic review by Rosen et al. identified only one case–control study that assessed mercury levels and pre-eclampsia. As they did not find a marked difference, an association was not assumed. However, the authors emphasized the large gap in knowledge [48]. In contrast, the review of Kahn and Trasande suggests that mercury may be of concern [51], though this was based uniquely on a prospective cohort study in which pregnant women with occupational exposure had higher mercury concentrations (*p* < 0.001) and an increased risk of pre-eclampsia (Relative Risk: 3.67; 95% CI 1.25 to 10.76) [52].

### 4.2. Preterm Birth

Preterm birth is defined as birth before 37 completed weeks of gestation and can occur spontaneously. In the cases of medically indicated preterm births, labour is initiated due to pregnancy complications. One example of maternal medical conditions leading to indicated preterm birth are hypertensive disorders of pregnancy, including pre-eclampsia. Other examples are chronic hypertension, restrictive lung diseases, and gestational diabetes mellitus [53]. Preterm birth is a major global health problem. The World Health Organisation estimates more than 10% of all live births worldwide to be preterm in 2010, and global inequalities are pronounced [54]. In 2010, the highest rates were estimated for South-eastern Asia, Southern Asia, and Sub-Saharan Africa with mean preterm rates between 12 and 14% [55].

The systematic reviews that performed a meta-analysis to examine the association between specific metals and preterm birth are listed in Table 1.

In a recent meta-analysis, no eligible studies conducted in South Asia were retrieved, and for African populations, only limited data are available to analyse the association between metal exposure and preterm birth [56]. Nonetheless, not only preterm birth rates but also the risk of metal exposure during pregnancy are likely to be high in the low- and middle-income countries in these regions [3]. Even in some high-income countries, the incidence of preterm birth is still high. For example, in the United States in 2020, just over 10% of all births were preterm [57]. In a recent meta-analysis, maternal exposure to metals was associated with moderate certainty and with an increased risk of preterm birth (OR 1.23; 95% CI, 1.17 to 1.29) [56]. In subgroup analyses, the pooled OR for studies with sample collection during early or mid-pregnancy (OR, 1.55; 95% CI, 1.27 to 1.88) was higher compared to when samples were collected in late pregnancy (OR, 1.17; 95% CI, 1.11 to 1.23) [56]. This observation was confirmed for cadmium in a further systematic review that found a higher increased risk of preterm birth with first-trimester cadmium exposure (Risk Ratio (RR), 3.06; 95% CI, 1.69 to 5.54) compared to the second (RR: 1.61; 95%CI: 1.12 to 2.32) and third (RR: 1.28; 95%CI: 0.93 to 1.78) trimester exposure estimates. However, the subgroups for the exposure measured during the first and second exposure consisted of only one study each [58], indicating the challenges of collecting data early in pregnancy.

**Table 1 ijerph-20-05536-t001:** Results from systematic reviews that performed a meta-analysis to assess the association between non-essential trace element exposure and the risk of preterm birth.

Exposure	Sample Type	Measure of Association	Heterogeneity		*n*	Reference
		Estimate (95% CI)	I^2^	*p*		
Arsenic	maternal serum maternal urine	OR, 1.06 (0.92 to 1.21)	57.7%	0.037	6	Wu et al., 2022 [56]
Cadmium	maternal serum maternal urine	OR, 1.33 (1.06 to 1.67)	82.0%	0.000	8	Wu et al., 2022 [56]
Cadmium	maternal blood maternal urine	RR, 1.32 (1.05 to 1.67)	90.0%	0.000	5	Amegah et al., 2021 [58]
Cadmium	maternal blood maternal serum maternal urine	OR, 1.32 (1.08 to 1.61)	81.4%	0.000	10	Asefi et al., 2020 [59]
Lead	cord blood maternal serum maternal urine	OR, 1.39 (1.10 to 1.76)	88.1%	0.000	10	Wu et al., 2022 [56]
Mercury	maternal serum maternal urine	OR, 1.09 (0.94 to 1.27)	32.7%	0.203	5	Wu et al., 2022 [56]

CI: confidence intervals; *n*: number of included studies; OR: odds ratio, RR: relative risk. Shading indicates statistical significance. Blue: statistically significant; bright orange: I^2^ > 50%: substantial heterogeneity, *p* < 0.05; dark orange: I^2^ > 75% considerable heterogeneity, *p* < 0.01.

Khanam, Kumar et al. published a scoping review examining the association between prenatal exposure of environmental metals such as lead, mercury, cadmium, and arsenic and the incidence of preterm birth. They included 20 papers on lead exposure, concluding that lead is a risk factor for preterm birth in populations that are exposed to generally high environmental lead levels. However, in this review, the results were conflicting in studies that examined the association of preterm birth with placental or cord blood lead levels. The authors speculate that potential reasons for a lack of association could be low sample sizes, methodological concerns, or low lead levels [3]. Elsewhere, Wu et al. found a correlation in their systematic review between maternal lead exposure and the risk for preterm birth (OR, 1.39; 95% CI, 1.10 to 1.76) [56].

The studies included by Khanam, Kumar et al. for the association between maternal cadmium exposure and preterm birth indicate that here, association may be context-specific. Three of the nine included studies found no association, and one study that measured placental cadmium levels observed a negative association. Three of the five papers that observed a positive association were conducted in China, where high cadmium levels had been measured. They concluded that only populations with high environmental cadmium exposures are affected [3]. However, not all studies that observed a high exposure found an association, including one study in Myanmar. They reported having observed higher urinary cadmium concentrations than in previous findings from China [60]. Wu et al. found a correlation (OR, 1.33; 95% CI, 1.06 to 1.67) between the risk for preterm birth and maternal cadmium exposure. However, there was high heterogeneity between studies in this meta-analysis, likely due to differences in maternal age at delivery, study setting, study design, ethnicity, and timing of sample collection [56]. Similarly, cadmium exposure resulted in a 32% increased risk of preterm birth (RR, 1.32; 95% CI, 1.05 to 1.67) in the meta-analysis by Amegah et al., which included five cohort studies. However, again, high heterogeneity was observed. The review authors performed a dose–response meta-analysis for three studies that provided estimates for different levels of cadmium exposure, which showed that an increase of 1 µg/L in cadmium exposure was associated with a 0.5% (OR, 1.005; 95% CI, 1.003 to 1.007) increase in the risk of preterm birth, and suggested a causal association [58]. A third meta-analysis included five additional studies with mixed study designs to assess the association between cadmium exposure and preterm birth. This review suggested that there might be a positive association (OR, 1.32; 95% CI, 1.08 to 1.61) but, again, highlighted the considerable heterogeneity within the included studies. They also performed a subgroup analysis for sample types (blood, serum, urine) but did not observe an association for any group, possibly due to the limited number of studies in the subgroups [59]. More recently, Flannery et al. conducted a scoping review including studies on cadmium exposure, where exposure mostly took place via dietary intake. Four studies assessed the association between maternal cadmium exposure and early or premature birth, including two large studies from China and one smaller study in Japan that found an association. A study in the United States, however, did not observe an increase in risk but involved lower median cadmium levels than the median of the two studies from China [61].

In a review focusing on the effects of chronic arsenic exposure, there were only three eligible studies assessing the association of arsenic exposure with preterm birth, of which two did not observe a statistically significant association [62]. Khanam et al. assessed articles that were published up to 2019 and also reported a sparsity of data on this relationship, and their review was inconclusive despite including 10 eligible studies [3]. Across six studies, Wu et al. found that the association between arsenic exposure and preterm birth did not reach significance (OR, 1.06; 95% CI, 0.92 to 1.21) [56].

For mercury, the findings by Khanam et al. were also inconsistent. Across seven included studies, five had a small sample size, and the study with the largest sample size (*n* = 362,625) was the only study that was not evaluated to be of high quality [3]. Wu et al. included five studies on this topic, but no statistically significant association of mercury with preterm birth was found (OR, 1.09; 95% CI, 0.94 to 1.27) [56].

### 4.3. Prenatal Growth

Commonly used anthropometric measures to assess prenatal growth include neonatal birth weight, birth length, and head circumference. The systematic reviews that performed a meta-analysis to examine the association between specific metals and measures of birth weight are listed in Table 2.

#### 4.3.1. Birth Weight

Several recent systematic reviews have examined the association with non-essential trace metal exposure and birth weight, as described in Table 2.

Across several reviews included in this manuscript, cadmium was associated with a lower birth weight. Khoshhali et al. found a weak association (Fisher-Z test, −0.04; 95% CI, −0.07 to −0.01) but reported high heterogeneity, mainly due to gestational age [64]. Huang et al. observed that a 50% increase in blood cadmium levels corresponded with an 11.57 g (95% CI, 4.30 to 18.85) decrease in birth weight, and a 50% increase in urine cadmium levels corresponded with a 6.15 g (95% CI, 1.49 to 10.81) decrease. Stratification for sex, however, revealed an association for female neonates only. Besides birth weight, the risk of low birth weight, which is defined as having a birth weight below 2500 g, was higher in the studies with elevated urinary cadmium (OR, 1.12; 95% CI, 1.03 to 1.22) but not in those with elevated blood cadmium (OR, 1.13; 95% CI, 0.74, 1.72) [65]. In five studies, Amegah et al. found an increased risk of low birth weight with cadmium exposure (RR, 1.21; 95% CI, 1.02 to 1.43), though again, there was a moderate heterogeneity. Despite the statistically significant association, significance was lost when a dose–response analysis was conducted (OR, 1.00; 95% CI, 0.998 to 1.001). The meta-analysis also included eight studies to assess the association with overall birth weight, which found that an incremental increase of 1 µg/L in urine and blood cadmium corresponded with a 42.11 g (95% CI, 15.18 to 69.03) reduction in birth weight. Again, there was substantial evidence of heterogeneity among the studies. Both the association with birth weight and the risk of low birth weight were more pronounced among female infants, but all estimates for the sexes separately were not statistically significant [58]. Finally, Flannery et al. included studies on maternal cadmium exposure, for which exposure took place mostly via dietary intake and which found a negative association with birth weight or low birth weight for at least one sex in 13 of the 20 included studies. Five studies also found a statistically significant or stronger association in female infants. However, among the nine studies that measured fetal exposure, six did not find a relationship with birth weight [61].

Besides low birth weight, small for gestational age (SGA) is also associated with a higher risk of mortality and morbidity. Major risk factors are previous SGA birth, diabetes, vascular diseases, chronic hypertension, and hypertensive disorders of pregnancy [69]. SGA is defined as having a birth weight below the 10th percentile when considering the gestational age (GA) and sex.

Habibian et al. indicated a direct, positive association between maternal blood cadmium exposure and risk of SGA (OR, 1.31; 95% CI, 1.16 to 1.47) [68], an association confirmed in the review of Amegah et al. (RR, 1.10; 95% CI, 0.96 to 1.27; 3 studies). However, as well as for low birth weight, in the fixed-effects dose–response regression analysis, no evidence was found of an association with cadmium exposure (OR, 1.025; 95% CI, 0.933 to 1.126) in this review. In contrast to low birth weight, however, the RR values for high and medium cadmium exposure differ. While medium cadmium exposure was associated with no risk (RR, 1.00; 95% CI, 0.73 to 1.37), high cadmium exposure was associated with SGA (RR, 1.49; 95% CI, 1.08 to 2.07) [58]. Howe et al. conducted an environmental mixture analysis of metal impacts on birth weight for gestational age (GA) z-score. Using a US reference, the authors pooled data from three cohorts conducted in the United States that measured seven metal concentrations in maternal urine samples. The results for cadmium were null for all three cohorts, and no overall association was found (0.02, 95% CI, −0.08 to 0.13), though the authors asserted that this result may contradict prior studies due to the relatively low cadmium levels [67].

Maternal arsenic exposure also seems to be associated with birth weight. Zhong et al. found an inverse association (β, −25.0 g; 95% CI, −41.0 to −9.0) but observed strong heterogeneity, and four of the twelve included studies did not find an association [63].

In the meta-analysis of Wang et al., reduced birth weight was associated with maternal and cord blood lead levels. Through converting standardized regression coefficients (Table 2) to metric regression coefficients, a 1 μg/dL increase in maternal blood and cord blood lead levels was associated with a 79.6 g and 22.2 g decrease in birth weight, respectively. However, when restricted to studies that reported adjusted effect estimates, the pooled association with maternal blood was weakened, and that with cord blood lost statistical significance [66].

Conversely, for mercury, Dack et al. found no strong evidence for an association with birth weight. However, included studies that were of high quality, as well as those that explored non-linearity or observed higher mean levels of mercury, more frequently observed a negative association. The authors state that this may suggest a threshold effect [70]. The recent mixture analysis by Howe et al. also observed no association in the cohort that had lower urine mercury levels and a narrower range in the samples. Across all three included cohorts, after accounting for co-exposure of six other metals in the mixture, a potential inverse linear association was identified between mercury urine levels and birth weight for GA but was not significant (−0.09; 95% CI, −0.20 to 0.03) [67].

#### 4.3.2. Birth Length and Head Circumference

Two other predictive measures of prenatal growth are birth length and birth head circumference. Related meta-analyses are listed in Table 3.

Regarding arsenic, the summary regression coefficient in the study of Zhong, Cui et al. indicated that arsenic exposure is associated with birth length (β, −0.12 cm; 95% CI, −0.17 to −0.07). Arsenic exposure was also associated with head circumference (β, −0.12 cm; 95% CI, −0.24 to −0.01). In both cases, the number of studies investigating the relationship was small [63].

Most studies included in the systematic review by Dack et al. did not find an association between mercury exposure and birth length. However, studies measuring mercury in placental tissue were mixed. The review also indicated that there is no association between mercury exposure and head circumference. However, this should be interpreted with caution because the results were mixed when only the high-quality studies were considered [70].

Cadmium exposure was also not associated with either birth length (0.01; 95% CI, 0.04 to 0.02; *n* = 11) or birth head circumference (0.02; 95% CI, 0.06 to 0.02; *n* = 8), according to a meta-analysis conducted by Khoshhali et al. [64], and supported by Flannery et al. [39]. Their scoping review suggested no relationship with birth length because most studies eligible for their review found no statistically significant relationship, namely, six out of seven studies measuring fetal cadmium exposure and seven out of eleven studies measuring maternal cadmium exposure. For head circumference, three of the four included studies also did not observe a significant association with fetal exposure. Of the eight studies that assessed maternal cadmium exposure, again two studies observed a negative association in female infants only [61]. Amegah et al. found no association between cadmium exposure and birth length (0.004 cm; 95% CI, −0.082 to 0.091). In contrast to the two other reviews, however, an association between head circumference and cadmium levels in blood and urine was revealed. Cadmium exposure, per 1 µg/L increment, was associated with a 0.105 (95% CI, 0.029 to 0.181) reduction in head circumference, with no evidence of heterogeneity observed [58].

## 5. Discussion

Over the last few decades, numerous studies have been published assessing the association between exposure to non-essential trace elements and pregnancy outcomes. These include systematic reviews that have collated and analysed mainly cohort and case–control studies. However, they often report serious limitations that point to the need for further, robustly designed studies, to assess the effect that non-essential trace elements may have on pregnancy outcomes. The effects observed in studies included in this review include pre-eclampsia and preterm birth, which were associated with cadmium and lead exposure, with cadmium, lead, and arsenic associated with lower birth weight. Arsenic was also associated with a lower birth length and head circumference at birth.

One of the biggest challenges to the conclusions of these systematic reviews is the small number of eligible studies, which often have a small sample size [2,48,62], thereby resulting in underpowered statistical tests [58,65] and possibly biased subgroup analyses [63]. Therefore, the results should be interpreted with caution.

Another considerable challenge in many reviews is that, among studies, different methods are used to assess exposure to non-essential trace elements [48,63,64,70], as visualised in Figure 1. As a result, comparison among studies is challenging. The restriction to studies with the same method reduces the sample size and the feasibility of meta-analyses [68]. Heterogeneity is often considerable, due to exposure indicators from different body tissues and fluids, and different time periods of exposure. Blood and urine samples are widely used to assess recent exposure [48,51,62,63,65], while hair, nails, and placental samples are used to examine long-term exposure [48,51,63,70]. Placental tissue might be most accurate in reflecting exposure throughout pregnancy [51,70], but its validity has also been questioned due to a lack of standardized methods for the collection and preservation of the placenta after delivery and of reported criteria for placental samples used in studies [15]. Standardized criteria for placental collection and assessment are required [11].

Another factor contributing to heterogeneity among studies is the difference in timing of the collection of samples in relation to the timing of delivery [1,56]. The uptake, mobilization, and excretion of the studied metals and, thus, their concentrations in body tissues and fluids may change throughout pregnancy, even if external exposure is constant. Therefore, the choice of the method and the timing of sampling are key to measuring the true association between exposure and outcome. One example of how the concentration of a metal can vary throughout pregnancy is the fact that during pregnancy, lead can be mobilized from maternal bone [71]. Based on this concept, Poropat et al. suggest that blood lead levels should be interpreted differently depending on when the samples have been taken, as blood lead levels measured before mid-gestation might not accurately reflect exposure during pregnancy [50]. This also raises the significance of periconceptional exposure. Body stores of metals can be mobilized during pregnancy and, therefore, might be an additional source of exposure for the mother and the foetus, as depicted in Figure 1. This suggests that periconceptional exposure to non-essential trace elements might also affect pregnancy outcomes. However, regarding the chosen elements and the specific pregnancy outcomes, no study was identified that examined this hypothesis. Only one study mentioned the periconception period, stating that samples taken during this period might help identify risk factors for hypertensive disorders [51]. A further example of how pregnancy can influence levels of metals is in arsenic metabolism. Throughout pregnancy, there is an increase in the efficiency of arsenic methylation, which is associated with improved urinary excretion relative to blood arsenic concentrations [72]. This suggests that pregnancy can cause changes in exposure levels, which should be considered in further studies to avoid misclassification bias. Taking several measurements throughout pregnancy may be important to better understand the interaction between different elements in the context of pregnancy.

To reduce heterogeneity, studies could speciate metals. For example, the intake of benign dietary forms of arsenic, such as from seafood, can lead to misinterpretation of the total arsenic levels [73]. Speciation analysis is necessary in clinical samples to fill remaining gaps in the knowledge of mechanisms of toxicity [2].

Most studies do not assess multiple exposures simultaneously, even though humans are usually exposed to a mixture of compounds. Mixture analyses would enable control for highly correlated metals and interactions [48]. Interactions between metals may also be assumed. However, a recently published review about prenatal exposure to metals and metalloids and birth weight for gestational age did not identify potential interactions between metal pairs. They focused on seven elements, namely, cadmium, cobalt, mercury, molybdenum, nickel, antimony, and tin. They pooled data from three cohorts in the United States and used the Bayesian Kernel Machine Regression (BKMR) to investigate associations [67]. Besides studies on interactions between non-essential elements, there are also papers suggesting interactions with essential elements [2]. A scoping review used a multipollutant approach to assess the association between exposure to metal mixtures and cardiovascular risk factors and outcomes. They included two studies focusing on pre-eclampsia. One reported an increased risk for individuals with higher scores for a principal component characterized by high loadings for cadmium, manganese, and lead, and who had scores below the median for a principal component that was characterised by high loading for copper, selenium, and zinc [74]. An example for interactions between non-essential and essential elements is provided by calcium. During pregnancy and lactation, lead can be released from the skeleton into the maternal circulation, potentially putting the infant at risk. Gulson and colleagues found that women who took a calcium supplement during pregnancy had lower blood lead concentrations in the third trimester compared to women whose dietary calcium intake was low, indicating that calcium supplementation may delay lead release from bone. Blood lead concentrations were higher postpartum in both the calcium-supplemented and low-calcium groups compared to other time points [71]. This points to a complex relationship between essential and non-essential trace metals and normal physiology at life stages of increased nutritional demand such as pregnancy. Another example is the association between cadmium and zinc. The results of a case–control study by Laine et al. suggest that zinc may reduce the risk of cadmium-associated disease during gestation [75]. In Bangladesh, Kippler et al. concluded that elevated placental cadmium concentration appears to result in impaired zinc transfer to the fetus [76]. More studies examining the interaction between different elements are required to fill the knowledge gaps in the etiology of pregnancy outcomes.

Another common limitation of reviews was the quality of the included studies. Reviews that compared studies of different quality observed that restriction to high-quality studies or studies that adjusted for potential confounders changed the results. Included studies often used different confounding variables or did not control for confounders at all [66,70]. Insufficient information about the subjects in studies, such as gender, ethnicity, or the timing of sampling, can render it challenging to perform subgroup analyses [68].

This narrative overview of recent systematic reviews and other literature syntheses demonstrates that exposure to non-essential, potentially harmful trace elements may be associated with pre-eclampsia, preterm birth, and birth size. However, these findings should be interpreted with caution since the reviews had some major limitations. The major strengths of our review are its summary of a range of evidence syntheses and that it is not limited to systematic reviews. However, it is limited by being mainly based on reviews published between 2017 and 2022 that focus on specific metals and metalloids only. More high-quality studies with a large sample size are required to confirm associations. Studies measuring exposure from before pregnancy until after pregnancy and using different exposure assessment methods could help further understand the mechanisms and etiology of the specific pregnancy outcomes for improving prevention of adverse pregnancy outcomes.

## Figures and Tables

**Figure 1 ijerph-20-05536-f001:**
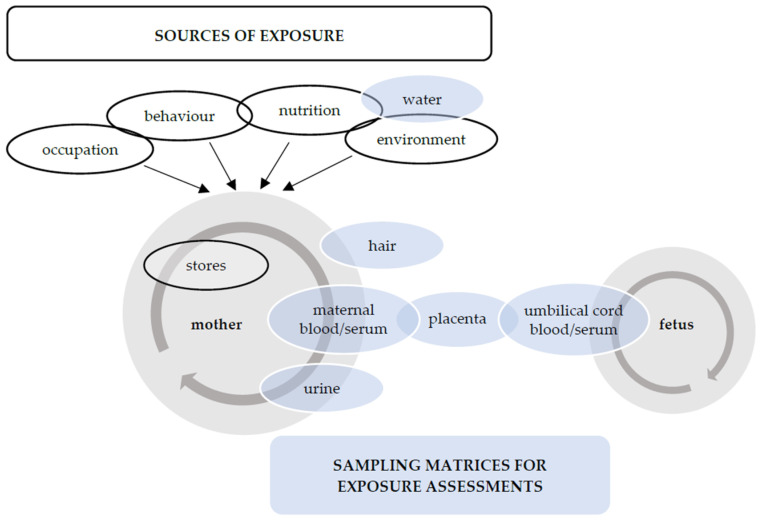
Categories of exposure sources discussed in this review and matrices that were included in the cited reviews for measuring non-essential trace element exposure. Eighteen of the reviews included in the present review listed included studies and exposure assessment methods. At least 3 of the 18 studies used the presented sampling matrices to assess exposure to the selected non-essential trace elements. The figure also shows the importance of using multiple sampling matrices to investigate toxico-kinetics in the placenta, which is a complex topic. The amount of accumulation and passage, as well as the involved mechanisms, differs among the heavy metals and their chemical forms [10]. There are inconsistencies and ongoing debate regarding this topic; however, it seems that cadmium accumulates in the placenta, and lead does not accumulate but rather crosses the placenta [10,11].

**Table 2 ijerph-20-05536-t002:** Systematic reviews that performed a meta-analysis to assess the association between non-essential trace element exposure and measures of birth weight.

Outcome	Exposure	Sample Type	Measure of Association		Heterogeneity		*n*	Model	Reference
			Effect Size	Effect Estimate (95% CI)	I^2^	*p*			
BW	Arsenic	maternal blood, maternal urine, maternal hair, drinking water	summary regression coefficient [g BW] in populations exposed to arsenic	−25.0 (−41.0 to −9.0)	73.3%	0.000	12	REM	Zhong et al., 2019 [63]
BW	Cadmium	cord blood, maternal blood, maternal urine, placenta	Fisher-Z for the studies obtained from sensitivity analysis	−0.04 (−0.07 to −0.01)	37.6%	0.024	18	FEM	Khoshhali et al., 2020 [64]
		maternal blood, maternal urine, cord blood	summary-effect size: difference in BW [g] per 1 µg/L increment in cadmium	−42.11 (−69.03 to −15.18)	64.6%	0.006	8	REM	Amegah et al., 2021 [58]
		maternal blood	linear regression coefficient: difference in BW [g] associated with 50% increase in cadmium	−11.57 (−18.85 to −4.30)	52.6%	0.077	5	REM	Huang et al., 2019 [65]
		maternal urine		−6.15 (−10.81 to −1.49)	0.0%	0.507	5	FEM	Huang et al., 2019 [65]
BW	Lead	cord blood	unadjusted standardized regression coefficients	−0.120 (−0.239 to −0.001)	62.5%	0.014	7	REM	Wang et al., 2020 [66]
			adjusted standardized regression coefficients	−0.017 (−0.045 to 0.012)	0%	0.838	4	REM	Wang et al., 2020 [66]
		maternal blood	unadjusted standardized regression coefficients	−0.094 (−0.157 to −0.030)	40.0%	0.101	9	REM	Wang et al., 2020 [66]
			adjusted standardized regression coefficients	−0.037 (−0.073 to −0.002)	67.1%	0.028	4	REM	Wang et al., 2020 [66]
BW for GA	Cadmium	maternal urine	SD difference in BW for GA for a 25th to 75th percentile change in cadmium	0.02 (−0.08 to 0.13)	-	-	3	FEM	Howe et al., 2022 [67]
BW for GA	Mercury	maternal urine	SD difference in BW for GA for a 25th to 75th percentile change in mercury	−0.09 (−0.20 to 0.03)	-	-	3	FEM	Howe et al., 2022 [67]
LBW	Cadmium	maternal blood	odds ratio	1.13 (0.74 to 1.72)	0%	0.629	2	FEM	Huang et al., 2019 [65]
		maternal blood, maternal urine	relative risk	1.21 (1.02 to 1.43)	43.0%	0.135	5	REM	Amegah et al., 2021 [58]
		maternal urine	odds ratio	1.12 (1.03 to 1.22)	6.9%	0.341	2	FEM	Huang et al., 2019 [65]
SGA	Cadmium	maternal blood	odds ratio	1.31 (1.16 to 1.47)	9%	0.358	6	FEM	Habibian et al., 2021 [68]
		maternal blood, maternal urine	relative risk	1.10 (0.96 to 1.27)	0%	0.464	3	REM	Amegah et al., 2021 [58]

BW: birth weight; CI: confidence intervals; FEM: fixed-effect model; GA: gestational age; LBW: low birth weight; *n*: number of included studies; REM: random-effects model; SGA: small for gestational age. Shading indicates statistical significance. Blue: statistically significant; bright orange: I^2^ > 50%: substantial heterogeneity, *p* < 0.05; dark orange: I^2^ > 75% considerable heterogeneity, *p* < 0.01.

**Table 3 ijerph-20-05536-t003:** Systematic reviews that performed a meta-analysis to assess the association between non-essential trace element exposure and birth length and head circumference.

Outcome	Exposure	Sample Type	Measure of Association		Heterogeneity		*n*	Model	Reference
			Effect Size	Effect Estimate (95% CI)	I^2^	*p*			
BL	Arsenic	maternal blood, maternal urine	summary regression coefficient [cm]	−0.12 (−0.17 to −0.07)	0.0%	0.917	5	FEM	Zhong et al., 2019 [63]
BL	Cadmium	maternal blood, maternal urine, cord blood	Fisher-Z	−0.03 (−0.07 to 0.01)	62.7%	0.001	11	REM	Khoshhali et al., 2020 [64]
		maternal blood, maternal urine, cord blood	Fisher-Z, result of sensitivity analysis	−0.01 (−0.04 to 0.02)	41.1%	0.054	10	REM	Khoshhali et al., 2020 [64]
		maternal blood, maternal urine, cord blood	summary-effect size: difference [cm] per 1 µg/L increment in blood/urine cadmium levels	0.00 (−0.08 to 0.09)	14.9%	0.318	4	REM	Amegah et al., 2021 [58]
HC	Arsenic	maternal blood, maternal urine	summary regression coefficient [cm]	−0.12 (−0.24 to −0.01)	59.6%	0.030	6	REM	Zhong et al., 2019 [63]
HC	Cadmium	maternal blood, maternal urine, cord blood	Fisher-Z	−0.04 (−0.09 to 0.01)	61.3%	0.003	8	REM	Khoshhali et al., 2020 [64]
			Fisher-Z, result of sensitivity analysis	−0.02 (−0.06 to 0.02)	45.2%	0.051	7	REM	Khoshhali et al., 2020 [64]
		maternal blood, maternal urine, cord blood	summary-effect size: difference [cm] per 1 µg/L increment in blood/urine cadmium levels	−0.11 (−0.18 to −0.03)	0.0%	0.802	4	REM	Amegah et al., 2021 [58]

BL: birth length; CI: confidence intervals; FEM: fixed-effect model; HC: head circumference; *n*: number of included studies; REM: random-effects model. Shading indicates statistical significance. Blue: statistically significant; bright orange: I^2^ > 50%: substantial heterogeneity, *p* < 0.05; dark orange: I^2^ > 75% considerable heterogeneity, *p* < 0.01.

## Data Availability

No new data were created or analyzed in this study. Data sharing is not applicable to this article.

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
