# Peer review of "Effects of Non-Essential “Toxic” Trace Elements on Pregnancy Outcomes: A Narrative Overview of Recent Literature Syntheses"

_ijerph, 2023, doi:10.3390/ijerph20085536_

Round 1

Reviewer 1 Report

This manuscript covers an important subject matter and provides a well-organized, overarching summary of measures of select metals and their effects on pregnancy and birth outcomes. However, the study question and gap that this manuscript aims to fill is not clear and there are many technical gaps and misuses of technical terms in paper.

Abstract and Introduction

-          Abstract they’re looking at pregnancy outcomes, however the outcomes on which they end up focusing are measures of gestational age, birth weight, and birth size

-          Introduction focuses on pregnancy outcomes and maternal deaths – not birth outcomes. There are also many important systematic reviews on heavy metals (the constituents on which the authors focus) and birth outcomes, so it’s not clear the gap that this article aims to fill:

o   Jaishankar M, Tseten T, Anbalagan N, Mathew BB, Beeregowda KN. Toxicity, mechanism and health effects of some heavy metals. Interdiscip Toxicol. 2014 Jun;7(2):60-72. doi: 10.2478/intox-2014-0009. Epub 2014 Nov 15. PMID: 26109881; PMCID: PMC4427717.

o   Rahman A, Kumarathasan P, Gomes J. Infant and mother related outcomes from exposure to metals with endocrine disrupting properties during pregnancy. 2016 Nov; 569: 1022-1031. https://doi.org/10.1016/j.scitotenv.2016.06.134

-          I find the use of “toxic” problematic – from a toxicological sense, toxicity is based on dose. I recommend that the authors characterize the metals on which they focus as simply “metals.”

-          After reading the manuscript in full, it is not clear whether the authors’ research question focuses on mechanism of action, toxicity, exposure assessment, or epidemiologic associations. This should be made clear in the introduction and expanded upon in the methods.

Methods

-          The authors provide some useful details about date of the literature search and search term, however, the methods are missing key details necessary in a scoping review:

o   Type of studies that they review (i.e., the definition of “human” studies)

o   Inclusion and exclusion criteria

o   Data extracted from each study

o   Study population and location of studies

-          What do the authors mean by “most” studies?

Results

-          The authors summarize literature on some outcomes that were not included in search phrases (e.g., preeclampsia), so it’s not clear how they were included.

-          The summary of each metal could be more comprehensive. For example, there are papers missing and it would be useful to describe data gaps and support some statements with additional citations.

Author Response

Please see attachment - many thanks

Reviewer 2 Report

Thank you for the opportunity to review the manuscript entitled Effects of non-essential (toxic) trace elements on pregnancy out-comes: a narrative overview of recent literature syntheses, by Maria Dettwiler and colleagues.

In this narrative overview of recent systematic reviews and other literature, syntheses demonstrate that toxic trace element exposure may be associated with preeclampsia, preterm birth, and size at birth.

I found the manuscript really interesting. The authors have addressed the main non-essential (toxic) trace element exposure with pregnancy outcomes.

I have only one minor comment.

In the abstract, please briefly include the results of the search in terms of how many papers were retrieved from the search.

Author Response

Please see attachment - many thanks

Reviewer 3 Report

This is a fine review of an important subject which has received not a great deal of attention.  The graphic on exposure sources is clear and valuable.

However, it is not clear from the graphic. If all the trace elements reviewed are equally likely to cross the placenta and that would be good information to summarize. It is contained in bits in the text but a short summary of that would be meaningful to the reader. 

The short reviews of the elements chosen for investigation is worthwhile for some readers not very familiar with toxicological epidemiology studies, but because the audience is likely to be familiar with these elements, it might be condensed with references to appropriate sources. 

The description of the systematic review is missing some important information.  Where were the searched terms searched for? Did they need to be found in the title, or the title and abstract, or anywhere in the text?

The search terms intended to discover outcomes similar to birth weight  seems to be missing LBW, SGA, IUGR. In this way the review might be missing some reports and this should be corrected. 

The tables are quite well done providing the most important information.

The reference to the Gulson paper seems misleading in that it seems that calcium supplementation leads to mobilization even if delayed relative to some other unspecified group. I think the results are more complex and point to a different role of Ca supplementation. A main finding in the Gulson paper is that women taking a calcium supplement had lower lead levels in the 3rd trimester. "...in the third trimester, the mean PbB level was significantly lower for women (n = 10) who took a calcium (Ca) supplement (PbB 1.6 µg/dL) than those whose Ca intake was low"  and,  "For women who took the supplement, post-partum PbB levels were significantly higher than those in the other periods (2.7 vs 1.4–1.6 µg/dL)." They did have higher levels postpartum but only compared to levels when they were taking the supplement. 

Overall more synthesis would be valuable. In many places the paper reads as a paper-by-paper review and in some places it seems that each paragraph is a separate paper review. 

Author Response

Please see attachment - many thanks

Round 2

Reviewer 3 Report

The paper is improved by your revision. It does make a contribution to the field.